# SCALABLE CONSTRAINED MULTI-AGENT REINFORCEMENT LEARNING VIA STATE AUGMENTATION AND CONSENSUS FOR SEPARABLE DYNAMICS

## ABSTRACT

We present a distributed approach for constrained Multi Agent Reinforcement Learning (MARL) which combines learning of policies with augmented state and distributed coordination of dual variables through consensus. Our method addresses a specific class of problems in which the agents have separable dynamics and local observations, but need to collectively satisfy constraints on global resources. The main technical contribution of the paper consists of the integration of constrained single agent RL (with state augmentation) in a multi-agent environment, through a distributed consensus over the Lagrange multipliers. This enables independent training of policies while maintaining coordination during execution. Unlike other centralized training with decentralized execution (CTDE) approaches that scale sub optimally with the number of agents, our method achieves a linear scaling both in training and execution by exploiting the separable structure of the problem. Each agent trains an augmented policy with local estimates of the global dual variables, and then coordinates through neighbor to neighbor communication on an undirected graph to reach consensus on constraint satisfaction. We show that, under mild connectivity assumptions, the agents obtain a bounded consensus error, ensuring a collective near-optimal behaviour. Experiments on demand response in smart grids show that our consensus mechanism is critical for feasibility: without it, the agents postpone demand indefinitely despite meeting consumption constraints.

## 1 INTRODUCTION

In recent years, reinforcement learning (RL) has achieved significant success in solving diverse and complex decision-making tasks (Brown & Sandholm, 2019; Orr & Dutta, 2023; Silver et al., 2017). Many of these successes involve multiple agents and can be characterized as multi-agent RL (MARL). Generally, MARL addresses a sequential problem where a set of autonomous agents make decisions and interact in a shared environment to maximize a reward. However, MARL problems can quickly become intractable as the number of agents increases, since the number of possible interactions and the space of possible states can grow exponentially in the number of agents. Moreover, as all agents navigate and learn simultaneously, the environment may become non-stationary, invalidating many of the single-agent RL assumptions. In realistic scenarios, conflicting objectives often need to be balanced to achieve satisfactory solutions. This issue is exacerbated when increasing the number of autonomous agents, whose specific goals are not commonly aligned. Finding optimal strategies in multi-agent systems (MAS) usually require at least some level of coordination and communication.

Our work addresses distributed systems where agents have separable dynamics but must coordinate to satisfy global operational constraints. While this assumption is restrictive compared to general MARL with coupled dynamics, it enables linear scaling in both training and execution, making our approach practical for hundreds or thousands of agents. The setting remains genuinely multi-agent as agents must coordinate through consensus to satisfy global constraints. This structure naturally arises in infrastructure management (e.g., building thermostats, EV chargers) where local controllers make independent decisions but must respect system-wide limits (e.g., power grid capacity). When agents share the same MDP, we only need to train one policy for all agents, significantly reducing complexity. The multi-agent coordination occurs through consensus on a dual variable during execution. Our Constrained MARL (CMARL) framework has each agent maximize a primary reward while adhering

to a global average constraint on a secondary reward, with the constraint acting as the coupling mechanism.

Agents communicate only with immediate neighbors in the network, reflecting realistic constraints where global broadcast is infeasible. Through local communication, agents share dual variables to achieve consensus dynamically. Communication is essential to ensure that agents, while operating independently, coordinate to satisfy the global constraint. We develop a novel CMARL algorithm and validate it on smart grid management (Dileep, 2020), optimizing energy distribution while satisfying operational constraints. Our experiments demonstrate scalability across different network configurations with varying complexity and agent heterogeneity. The key contributions are:

1. A distributed CMARL algorithm ensuring consensus and constraint satisfaction over extended periods.

2. Scalable policy training through problem factorization based on state distributions.

3. Experimental validation on smart grid economic dispatch problems.

## 2 RELATED WORK

**Constrained Reinforcement Learning.** Our work builds upon CRL using Lagrangian multipliers (Altman, 2021; Borkar, 2005) and state-augmentation (Calvo-Fullana et al., 2023). We distinguish between safe RL that ensures per-step constraint satisfaction (Achiam & Amodei, 2019; Achiam et al., 2017; Chow et al., 2019), including recent safe MARL methods (Lu et al., 2021; Zhang et al., 2024), and average constraint satisfaction (ours) (Liang et al., 2018; Paternain et al., 2022). While safe RL is crucial for safety-critical applications, average satisfaction is more appropriate for resource management where temporary violations are acceptable if long-term consumption stays within bounds.

**Cooperative MARL** Methods like QMIX (Rashid et al., 2018), MADDPG (Lowe et al., 2017), and CTDE (Kraemer & Banerjee, 2016) scale poorly due to exponential growth in joint state-action spaces. Our approach trains policies independently, coordinating only through dual variable consensus.

**Constrained MARL.** Centralized Lagrangian approaches (Ding et al., 2023) are powerful but challenging to extend to CMARL (Agorio et al., 2024; Chen et al., 2024). Our approach combines single-agent training with distributed consensus, inheriting single-agent scalability while addressing global constraints.

**Decentralized MARL.** Recent work includes decentralized actor-critic (Chen et al., 2024) and multi-agent PPO (Mai et al., 2024) for demand response, but these lack explicit constraint handling. Independent learning like IPPO (Yu et al., 2022) scales well but ignores collective constraints. Our method adds primal-dual consensus for constraint satisfaction.

**Decentralized Constrained MARL.** Lu et al. (2021) propose Safe Dec-PG for distributed CMDPs where safety constraints involve all agents' joint actions, requiring peer-to-peer communication to coordinate. Zhang et al. (2024) develop Scal-MAPPO-L using local policy optimization with $k$-hop policies to handle global coupling from safety constraints, though they acknowledge exponential state-action space growth as a limitation. Importantly, both methods focus on "safety constraints" requiring per-step satisfaction, appropriate for safety-critical applications but unnecessarily restrictive for resource management where average satisfaction suffices. In contrast, we target average constraint satisfaction rather than per-step safety, and assume separable dynamics to achieve linear scaling. We also incorporate state-augmentation as in Calvo-Fullana et al. (2023) for immediate constraint response. This makes our method suitable for large-scale infrastructure management where coupling occurs through resource constraints rather than state interactions.

## 3 PROBLEM FORMULATION

Typically, CMARL is studied using the Markov Games framework (Littman, 1994), an extension of game theory to environments where the dynamics can be modeled using a Markov Decision Process (MDP). Markov games model interactions among multiple agents whose decisions influence a shared environment. In our distributed constrained setting, the Markov game is defined by the tuple

$\langle N, \{S^i\}_{i=1}^N, \{A^i\}_{i=1}^N, \{P^i\}_{i=1}^N, \{r_0^i\}_{i=1}^N, \{r_1^i\}_{i=1}^N \rangle$, where $N$ is the number of agents, $S^i \subset \mathbb{R}^m$ and $A^i \subset \mathbb{R}^d$ are compact sets denoting the states and actions of agent $i$, with $S := S^1 \times \cdots \times S^N$ and $A := A^1 \times \cdots \times A^N$ denoting the sets of joint states and actions. The joint state transition probability is given by $P : S \times A \to \Delta(S)$, with each individual agent's transition given by $P^i : S^i \times A^i \to \Delta(S^i)$, where $\Delta(S)$ is the probability simplex on $S$. We further denote by $r_0^i : S^i \times A^i \to \mathbb{R}$ the reward function for the main objective of agent $i$, and by $r_1^i : S^i \times A^i \to \mathbb{R}$ the reward function for the secondary objective subject to a constraint, with global counterparts defined as $r_0 : S \times A \to \mathbb{R}$ and $r_1 : S \times A \to \mathbb{R}$. At time $t$, given a joint state $s_t = (s_t^1, \ldots, s_t^N)$ and action $a_t = (a_t^1, \ldots, a_t^N)$, the system transitions to a new state $s_{t+1} = (s_{t+1}^1, \ldots, s_{t+1}^N)$ with probability $P(s_{t+1}|s_t, a_t)$. The Markov property ensures that the system dynamics only depend on the last state and action, i.e. $P(s_{t+1}|s_0, a_0, \ldots, s_t, a_t) = P(s_{t+1}|s_t, a_t)$. We consider a scenario with conflicting rewards, with $r_0$ acting as the main objective and $r_1$ as the secondary objective. Specifically, we aim to maximize the long-term average rewards for $r_0(s_t, a_t)$, while ensuring that the long-term average rewards for $r_1(s_t, a_t)$ exceeds a given threshold $c$.[1] This constrained optimization problem can be expressed as

$$\underset{\pi}{\text{maximize}} \quad \lim_{T \to \infty} \frac{1}{T} \mathbb{E}_{s,a \sim \pi} \left[ \sum_{t=0}^T r_0(s_t, a_t) \right] \tag{1a}$$

$$\text{subject to} \quad \lim_{T \to \infty} \frac{1}{T} \mathbb{E}_{s,a \sim \pi} \left[ \sum_{t=0}^T r_1(s_t, a_t) \right] \geq c. \tag{1b}$$

This is a multi-agent centralized problem, which is often impractical due to its poor scalability. Specifically, we are interested in problems that can be decomposed into distributed problems. Formally, we consider scenarios satisfying the following assumptions.

**Assumption 3.1** (Independent policies). Each agent $i$ selects an action taking into account only its own local state. Namely, $\pi(a_t|s_t) = \prod_{n=1}^N \pi^n(a_t^n|s_t^n)$.

**Assumption 3.2** (Separable dynamics). The actions of one agent do not affect the states of others. That is, state transitions are given by $P(s_{t+1}|s_t, a_t) = \prod_{i=1}^N P^i(s_{t+1}^i|s_t^i, a_t^i)$.

**Assumption 3.3** (Summable rewards). The global reward can be decomposed as the sum of individual rewards, i.e. $r_0(s_t, a_t) = \sum_{n=1}^N r_0^n(s_t^i, a_t^i)$ and $r_1(s_t, a_t) = \sum_{n=1}^N r_1^n(s_t^i, a_t^i)$.

*Remark* 3.4 (Scope and Limitations). Assumptions 3.1-3.3 significantly restrict the class of problems we address. Under these assumptions, agents do not influence each other's states or rewards directly, which excludes many classical MARL scenarios like multi-robot coordination or competitive games. However, these assumptions are satisfied in important real-world domains:

- **Smart Grid Management**: Buildings independently control their energy consumption but share grid capacity constraints

- **Distributed Computing**: Processes independently execute but share memory/bandwidth limits

- **Traffic Flow Control**: Vehicles follow independent routes but collectively impact road utilization

For problems with coupled dynamics, methods like those of Lu et al. (2021) and Zhang et al. (2024) are more appropriate, albeit at higher computational cost.

The first assumption allows each agent to operate based solely on local information, the second assumption ensures that the interactions of the agents are structured in a non-interfering manner, and the the third assumption ensures that global objectives can be achieved through local decisions. This

---

[1] For simplicity, we restrict ourselves to the single constraint case, though the results generalize to multiple constraints.

set of assumptions allows for the problem to be rewritten in the following form:

$$\max_{\pi^1,\ldots,\pi^N} \sum_{i=1}^{N} \lim_{T\to\infty} \frac{1}{T} \mathbb{E}_{s^i,a^i\sim\pi^i} \left[ \sum_{t=0}^{T} r_0^i(s_t^i, a_t^i) \right] \tag{2a}$$

$$\text{s.t.} \sum_{i=1}^{N} \lim_{T\to\infty} \frac{1}{T} \mathbb{E}_{s^i,a^i\sim\pi^i} \left[ \sum_{t=0}^{T} r_1^i(s_t^i, a_t^i) \right] \geq c. \tag{2b}$$

By defining value functions as the long-term average of each reward,

$$V_j^i(\pi^i) \triangleq \lim_{T\to\infty} \frac{1}{T} \mathbb{E}_{s^i,a^i\sim\pi^i} \left[ \sum_{t=0}^{T} r_j^i(s_t^i, a_t^i) \right], \tag{3}$$

we can then rewrite the maximization problem in (2) in the following more concise manner:

$$\underset{\pi^1,\ldots,\pi^N}{\text{maximize}} \sum_{i=1}^{N} V_0^i(\pi^i) \text{ subject to } \sum_{i=1}^{N} V_1^i(\pi^i) \geq c. \tag{4}$$

The resulting formulation now exhibits a certain degree of separability across agents, with each agent maximizing its own policy with respect to its individual value function, while still being coupled to the other agents through the global constraint. While the separable structure might suggest independent single-agent solutions would suffice, the global constraint in (4) creates a critical coordination challenge: without communication, agents cannot determine appropriate individual contributions to satisfy the collective constraint. This necessitates our consensus mechanism to coordinate the dual variables that encode constraint violation feedback.

## 4 METHODOLOGY

We begin by formulating the Lagrangian of the optimization problem in (4). This involves introducing Lagrange multipliers to transform the constrained optimization problem into a form where the constraints are incorporated into the objective function as penalty terms. Namely,

$$\mathcal{L}(\pi, \lambda) = \sum_{i=1}^{N} V_0^i(\pi^i) + \lambda \left( \sum_{i=1}^{N} V_1^i(\pi^i) - c \right) = \sum_{i=1}^{N} \left( V_0^i(\pi^i) + \lambda \left( V_1^i(\pi^i) - \frac{c}{N} \right) \right), \tag{5}$$

where $\lambda \in \mathbb{R}^+$ is the Lagrange multiplier (dual variable) for the inequality constraint. We rewrite the Lagrangian as individual agent components to maintain distributed formulation. The dual problem becomes

$$\underset{\lambda}{\text{minimize}} \left[ \sum_{i=1}^{N} \max_{\pi^i} \left( V_0^i(\pi^i) + \lambda \left( V_1^i(\pi^i) - \frac{c}{N} \right) \right) \right] \tag{6}$$

where summation and maximization are exchanged due to Assumptions 3.1 and 3.2. This decomposition enables independent local optimization while satisfying the global constraint. The problem exhibits strong duality (Paternain et al., 2019), so the optimal solution of (4) equals the saddle-point of (6).

*Remark* 4.1 (Comparison with Standard Primal-Dual Methods). Standard distributed primal-dual methods (Yarmoshik et al., 2024; Wang et al., 2024) require strongly convex objectives or extensive message passing. Our approach differs by: (i) using state augmentation from single-agent CRL (Calvo-Fullana et al., 2023) for non-convex policy optimization, and (ii) requiring only single-scalar neighbor communication. Integrating standard consensus (Xiao & Boyd, 2003) with state-augmented RL policies enables our scalability.

### 4.1 OFFLINE INDEPENDENT TRAINING

The primal step of (6) (policy learning) is distributable. For given $\lambda$, the problem decomposes across agents. Defining weighted reward $r_\lambda^i(s_t^i, a_t^i) \triangleq r_0^i(s_t^i, a_t^i) + \lambda r_1^i(s_t^i, a_t^i)$, the maximization becomes

$$\{\pi_\star^i(\lambda)\} = \underset{\pi^1,\ldots,\pi^N}{\arg\max} \sum_{i=1}^{N} \lim_{T\to\infty} \frac{1}{T} \mathbb{E}_{s^i,a^i\sim\pi^i} \left[ \sum_{t=0}^{T} r_\lambda^i(s_t^i, a_t^i) \right]. \tag{7}$$

Each agent's primal step follows standard unconstrained RL. However, standard dual methods can fail to produce feasible policies for CRL (Calvo-Fullana et al., 2023). We thus learn state-augmented policies $\pi^i(\lambda)$ in augmented space $S^i \times \mathbb{R}_+$ that maximize (5), instead of ordinary policies in $S^i$. Agents independently train these policies using any standard RL method. The trained policies handle any constraint level $c$ when coupled with our dual update mechanism.

## 4.2 ONLINE DUAL CONSENSUS

Determining $\lambda$ remains challenging since gradient descent on (5) couples all agents. Consider agents communicating over undirected graph $G = (V, E)$, where $V$ are vertices (agents) and $E \subset N \times N$ are edges. The neighborhood $\mathcal{N}^i = \{j \in V \mid (i, j) \in E\}$ contains nodes directly connected to $i$. We rewrite (6) in distributed dual consensus form:

$$\underset{\lambda^1,\ldots,\lambda^N}{\text{minimize}} \sum_{i=1}^{N} \max_{\pi^i} \left( V_0^i(\pi^i) + \lambda^i \left( V_1^i(\pi^i) - \frac{c}{N} \right) \right) \tag{8a}$$

$$\text{subject to} \quad \lambda^i = \frac{1}{|\mathcal{N}_i|} \sum_{n \in \mathcal{N}^i} \lambda^n, \quad i = 1, \ldots, N. \tag{8b}$$

The solution to (8) equals that of (6). Each agent holds local copy $\lambda^i$, with neighborhood constraints ensuring consensus. Using optimal policies from (7), we obtain

$$\underset{\lambda^1,\ldots,\lambda^N}{\text{minimize}} \sum_{i=1}^{N} \left[ V_0^i\big(\pi_\star^i(\lambda^i)\big) + \lambda^i \left( V_1^i\big(\pi_\star^i(\lambda^i)\big) - \frac{c}{N} \right) \right] \tag{9a}$$

$$\text{subject to} \quad \lambda^i = \frac{1}{|\mathcal{N}_i|} \sum_{n \in \mathcal{N}^i} \lambda^n, \quad i = 1, \ldots, N. \tag{9b}$$

## 4.3 PRIMAL-CONSENSUS UPDATE

To solve (9), each agent $i$ maintains $\lambda^i$ and iteratively (i) performs local gradient updates for constraint satisfaction and (ii) averages with neighbors' variables. With gradient step size $\alpha > 0$ and consensus step size $\epsilon > 0$, agent $i$ updates:

$$\lambda_{k+1}^i = \lambda_k^i - \alpha \nabla_{\lambda^i}\left[ V_0^i\big(\pi_\star^i(\lambda_k^i)\big) + \lambda_k^i \left( V_1^i\big(\pi_\star^i(\lambda_k^i)\big) - \frac{c}{N} \right) \right] - \epsilon \left( \lambda_k^i - \overline{\lambda}_k^i \right), \tag{10}$$

where $\overline{\lambda}_k^i = \sum_{n \in \mathcal{N}_i} \lambda_k^n / |\mathcal{N}_i|$ is the neighbor average. The first term performs local gradient descent; the second enforces consensus. These corrections drive all $\lambda^i$ to converge, matching the solution of (6).

*Remark* 4.2 (Relation to Centralized Dual). As agents optimize local $\lambda^i$ while enforcing neighbor consensus, $\{\lambda^i\}$ converges to the same value as the global $\lambda$ in (6). Thus, (10) provides fully distributed solution without centralized coordination.

# 5 ALGORITHM

Each agent $i$ optimizes its local policy by maximizing the Lagrangian given its current copy of the dual variable $\lambda^i$. To ensure that the policy appropriately accounts for constraint satisfaction, we augment each agent's state space with the local multiplier $\lambda^i$. This augmentation yields a policy $\pi_\star^i(s_t^i, \lambda_t^i)$ that views $\lambda^i$ as part of the state, so that standard reinforcement-learning (RL) algorithms can be used to learn this policy.[2]

If we have an optimal policy for a given set of multipliers $\pi_\star(s, \lambda)$, and we *continuously update* these multipliers (via 10), then the state-action trajectories generated by each agent satisfy the constraints in (2) (Calvo-Fullana et al., 2023, Theorem 1). Combining these ideas, we summarize the execution in Algorithm 1.

---

[2]In practice, many RL methods—e.g., policy gradient, value-based methods—can be adapted to handle such an augmented state.

**Theorem 5.1.** *Suppose the local value functions satisfy*

$$\left\| V_1^i\left(\pi_\star^i(\lambda_k^i)\right) - \frac{1}{N}\sum_{j=1}^{N} V_1^j\left(\pi_\star^j(\lambda_k^j)\right) \right\| \le \sigma, \tag{11}$$

*and let $w^i = |\mathcal{N}^i|/\sum_{j=1}^{N}|\mathcal{N}^j|$. Under mild conditions on the connectivity and step sizes, the execution of Algorithm 1 results in a bounded consensus error:*

$$\lim_{k\to\infty}\left\| \lambda_{k+1} - \sum_{i=1}^{N} w^i\,\lambda_k^i \right\| \le \frac{\rho^{\mathscr{L}}}{1-\rho^{\mathscr{L}}}\,\alpha\,\sigma, \tag{12}$$

*where $\rho$ and $\mathscr{L}$ relate to the graph's spectral properties and the number of communication steps per iteration (or partial consensus steps).*

Theorem 5.1 thus guarantees that all $\lambda^i$ stay close to each other, ensuring the distributed solution remains near the centralized optimum (and that the global constraints are met) even as policies are updated locally. The proof of the theorem appears in Appendix B.1.

This result highlights the relationship between the number of consensus iterations, $\mathscr{L}$, and the overall consensus error in the execution of Algorithm 1. The bound in Theorem 5.1 decreases as $\mathscr{L}$ increases, indicating that additional consensus steps reduce the discrepancy among agents' multipliers. In practice, a small $\rho$ (which occurs in well connected graphs) accelerates convergence, allowing $\mathscr{L}$ to remain small. For many real-world network structures, a single consensus iteration ($\mathscr{L}=1$) per gradient step is sufficient to ensure that the discrepancy in $\lambda^i$ remains below an acceptable threshold, minimizing communication overhead while maintaining effective coordination.

For every fixed multiplier $\lambda$, the policy $\pi_\star(\lambda)$ is *defined* as a maximizer of the inner problem $\max_\pi \mathcal{L}(\pi, \lambda)$. By Danskin's theorem, the gradient of this maximized objective with respect to $\lambda$ depends only on the partial derivative of $\mathcal{L}$, evaluated at the maximizer. Hence, in the multiplier update (Lines 6–8 of Algorithm 1) we treat $\pi_\star$ as constant without loss of correctness. This argument is standard in Lagrangian-based constrained RL (see, e.g., Calvo-Fullana et al., 2023)

---

**Algorithm 1** Distributed multiplier update with Separated Consensus and Gradient Steps

---

1: **Input:** Trained policies $\pi_\star^i(\lambda)$, learning rates $\alpha, \epsilon$, requirement $c$, number of consensus steps $\mathscr{L}$
2: **Output:** Trajectories satisfying the constraints
3: **Initialize:** Dual variables $\lambda_0^i = 0$, $\mu_0^i = 0$ for $i = 1, \ldots, N$
4: **for** $k = 0, 1, \ldots, K-1$ **do**
5:    **Gradient Descent Step:**
6:    $\lambda_{k+\frac{1}{2}}^i = \left[\lambda_k^i - \alpha\left(\frac{c}{N} - V_{1,k}^i\right)\right]_+$
7:    **Initialize Consensus Variable:**
8:    $\lambda_{\ell=0}^i = \lambda_{k+\frac{1}{2}}^i$
9:    **for** $\ell = 0, \ldots, \mathscr{L}-1$ **do**
10:      **Consensus Update:**
11:      $\lambda_{\ell+1}^i = \lambda_\ell^i - \epsilon\left(\lambda_\ell^i - \frac{1}{|\mathcal{N}_i|}\sum_{j\in\mathcal{N}_i}\lambda_\ell^j\right)$
12:    **end for**
13:    **Update for Next Iteration:**
14:    $\lambda_{k+1}^i = \lambda_{\ell=\mathscr{L}}^i$
15: **end for**

---

In practice, we perform only one consensus iteration per time step ($\mathscr{L}=1$).

## 6 USE CASE: SMART GRID MANAGEMENT

We apply our method to Demand Response (DR) in a district of buildings with solar energy and battery storage. Our goal is to minimize energy costs for each building while avoiding critical grid peaks through energy storage and load shifting. Each building's agent observes the current demand,

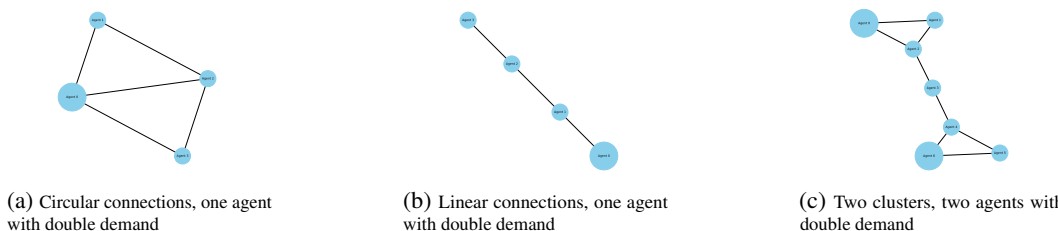

(a) Circular connections, one agent with double demand

(b) Linear connections, one agent with double demand

(c) Two clusters, two agents with double demand

Figure 1: Different communication networks and agent demands.

battery charge, and grid price, then decides how to allocate energy between grid and battery sources. The local objective is defined as $r_0^i(s_t^i, a_t^i) = -e_{\text{grid}}(s_t^i, a_t^i)\, p_t$, where $e_{\text{grid}}(s_t^i, a_t^i)$ is the building's grid consumption and $p_t$ is the energy price at time $t$. By maximizing $r_0^i$, agents minimize their grid electricity spending while respecting global consumption constraints.

The secondary reward $r_1^i(s_t^i, a_t^i) = e_{\text{grid}}(s_t^i, a_t^i)$ with constraint $\sum_{i=1}^N V_1^i(\pi^i) \leq c$ ensures that average total grid usage stays below threshold $c$ (a percentage of peak demand), maintaining grid stability. Agents can postpone unmet demand for later, and batteries charge automatically from solar generation. To ensure all demand is eventually met, we add a local constraint with reward

$$r_2^i(s_t^i, a_t^i) = d_t^i - e_{\text{grid}}(s_t^i, a_t^i) - e_{\text{bat}}(s_t^i, a_t^i), \tag{13}$$

where $d_t^i$ is the demand of agent $i$ at time $t$ and $e_{\text{bat}}(s_t^i, a_t^i)$ is the battery-delivered energy. Let $V_2^i(\pi^i)$ be the corresponding value function, defined as in (3). We then impose the local constraint

$$V_2^i(\pi^i) = 0,$$

which ensures that, in expectation, all of agent $i$'s demand is met over the long run. Since the constraint is local, it only affects the optimization problem of agent $i$. The training of the policy is performed following the state augmented procedure described in Section 4.1 and the updates of the global constraint and the consensus multipliers are performed as shown in Algorithm 1. For the handling of the local constraint we just add another term to the Lagrangian which only needs the addition of the following update

$$\nu_{k+1}^i = \nu_k^i - \eta\left(d_k^i - e_{\text{grid}}(s_k^i, a_k^i) - e_{\text{bat}}(s_k^i, a_k^i)\right), \tag{14}$$

with step size $\eta$. Energy prices, demand, and solar generation data come from City Learn (Vazquez-Canteli et al., 2020; Vázquez-Canteli et al., 2019).

# 7 EXPERIMENTAL RESULTS

We test our method[3] on the network configurations in Figure 1, which vary in connectivity and demand diversity. Less-connected networks challenge consensus, while heterogeneous demands create problems that require coordination to solve. We focus on the configuration in Figure 1c: two weakly connected groups where one agent in each has double the demand of others. Using PPO, we train just two policies—one for normal demand, one for double demand—demonstrating the efficiency of single-agent training with multi-agent execution. Individual Lagrange multipliers ($\lambda^i$) enable coordination during execution, with consensus being critical for linking training to execution and ensuring constraint satisfaction. Training uses 10,000 episodes of 80 timesteps each (about 3 days of demand), with multipliers sampled from $\lambda \in [0, 15]$ and $\nu \in [-20, 20]$. The dataset contains 3,000 hours of data with random episode starting points.

**Experimental Scope.** We focus on smart grid management as it naturally fits our structural assumptions while remaining complex enough to demonstrate consensus necessity. While broader evaluation would strengthen our claims, our primary contribution is demonstrating that state augmentation with

---

[3]All experiments were carried out on a MacBook Pro M3 with 8 GB RAM.

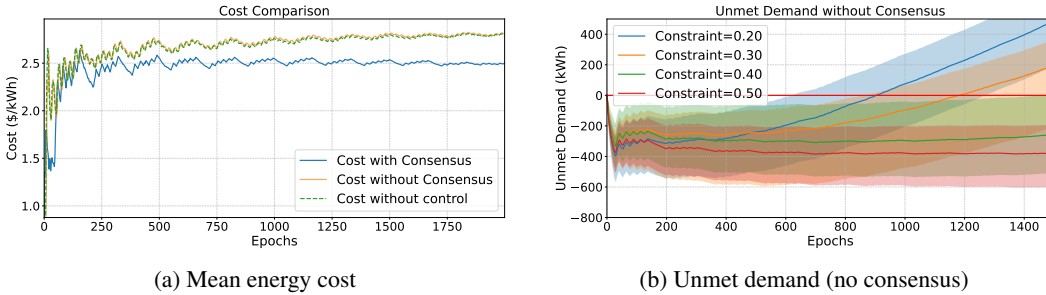

(a) Mean energy cost                (b) Unmet demand (no consensus)

Figure 3: Cost and demand-satisfaction trajectories for the consensus study. The dashed horizontal line in (b) marks zero unmet demand.

consensus enables unprecedented scalability, validated by scaling to 1000 agents (Figure A.2a), far beyond CTDE capabilities.

We set the constraint $c$ to 27% of peak demand (challenging yet feasible). Agents run for 3,000 timesteps with continuous multiplier updates (Algorithm 1). To demonstrate coordination importance, we compare two variants: with *consensus*, agents exchange multipliers $\lambda^i$ with neighbors and average them (lines 8–13 in Algorithm 1); without *consensus*, agents only perform local gradient updates without coordination. While both maintain grid consumption below the threshold (Figure 2a), the no-consensus version achieves this by indefinitely postponing demand rather than finding a true solution.

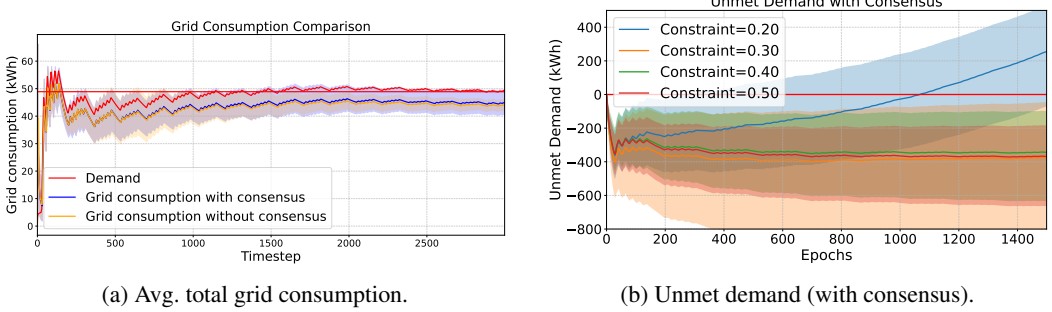

(a) Avg. total grid consumption.          (b) Unmet demand (with consensus).

Figure 2: Effect of consensus on grid consumption and unmet demand under different constraint levels.

Testing four constraint levels $c \in \{0.2, 0.3, 0.4, 0.5\}$, we find that consensus achieves stable unmet demand for feasible cases ($c \geq 0.3$), with lower constraints allowing more grid usage as expected (Figure 2b). At $c = 0.2$, even consensus cannot find a solution. Without consensus, the problem becomes infeasible even for moderate constraints like $c \in \{0.3, 0.4\}$ (Figure 3b). Crucially, at our target $c = 27\%$, the no-consensus version fails to solve the problem despite meeting grid constraints—its multipliers never converge (Figure 4b), preventing optimal solution discovery.

The absence of consensus produces both higher operating cost (Figure 3a) and a continued growth of deferred demand (Figure 3b). These complementary views underline the practical value of the lightweight neighbour-to-neighbour communication adopted in Algorithm 1.

Figure 4a shows that exchanging the current $\lambda^i$ with immediate neighbours causes convergence to the same value, thereby satisfying the global grid-consumption constraint. Without this exchange (Figure 4b) the two high-demand agents push their multipliers to the hard cap of 15, signalling that dual ascent has saturated before a feasible primal solution was found.

Because our algorithm is explicitly designed for constrained optimization whereas most state-of-the-art MARL methods are not, we ran a second experiment for which we searched for static penalty weights that make the task solvable at all. We trained a grid of Independent PPO (IPPO) agents, one

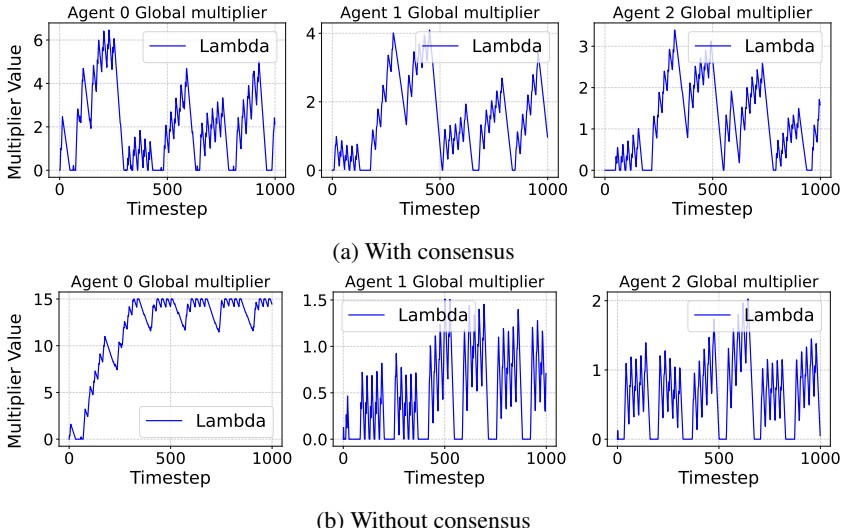

(a) With consensus

(b) Without consensus

Figure 4: Evolution of the global multipliers $\lambda^i$ for the first three agents during execution.

for every $(\lambda, \nu) \in [0, 15] \times [0, 28]$, yielding 414 models in total. Only eight pairs satisfy the 27 % grid-consumption limit and prevent postponed demand from diverging; they are listed in Table 1 and visualized in Figure A.1. In our comparisons, we provide the baselines with one of those pairs to ensure a fair (if slightly biased in their favour) comparison. From the eight feasible pairs we selected $(\lambda^\star, \nu^\star) = (8, -8)$ and re-trained four multi-agent baselines: MAPPO, MADDPG, MASAC, and ISAC. Each baseline therefore ran with fixed penalty weights, whereas our method continued to adapt the multipliers online. For every algorithm we executed 10 roll-outs and recorded (i) the trajectory that came closest to satisfying the 27 % grid threshold and (ii) the mean total cost of this "best" run. Figure A.3 (Appendix A) shows that only MAPPO and ISAC keep grid consumption near the limit, while MASAC and MADDPG overshoot. Even these two "successful" baselines closely match the operating cost of our decentralized multiplier-adaptive method, which in turn achieves 0 % infeasible roll-outs across all seeds. Despite their hand-tuned advantage, the baselines still violate the grid constraints, highlighting their sensitivity to the fixed choice $(\lambda^\star, \nu^\star)$. All four baselines rely on centralized components; MAPPO, MASAC and ISAC use a joint critic, while MADDPG conditions each critic on the full joint action.[4] Consequently, their computational and memory costs explode as the population grows, limiting practical use to a few dozen agents. Our method keeps both training and execution fully decentralized, needs only one policy per agent type, and scales linearly in the number of agents (see Figure A.2a), giving it a clear advantage for large systems.

## 8 CONCLUSION

We presented a distributed approach to constrained MARL that combines state-augmented policy learning with consensus-based coordination. While our assumptions of separable dynamics and summable rewards restrict applicability compared to general constrained MARL methods, they allow a highly scalable solution for an important class of real world problems. Our key contributions are: (i) extending single-agent state augmentation to multi-agent settings through distributed consensus, (ii) proving convergence bounds for the consensus error, and (iii) demonstrating linear scaling to thousands of agents.

---

[4]The policies may execute decentralized actions at test time, but training still scales at least quadratically with the number of agents because the critics ingest the joint state–action tuple.

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

# A ADDITIONAL EXPERIMENTAL FIGURES

This appendix expands the results of Section 7 with the complete diagnostic curves, ablations, and heat-maps that underpin the summary metrics shown in the main text. Unless stated otherwise, all experiments rely on the network of $n = 7$ agents sketched in Figure 1c, where two agents have twice the demand of the remaining five. We denote the global and local penalty multipliers by $(\lambda, \nu)$ to match the notation used in Section 7.

## A.1 GRID SEARCH OVER FIXED MULTIPLIERS

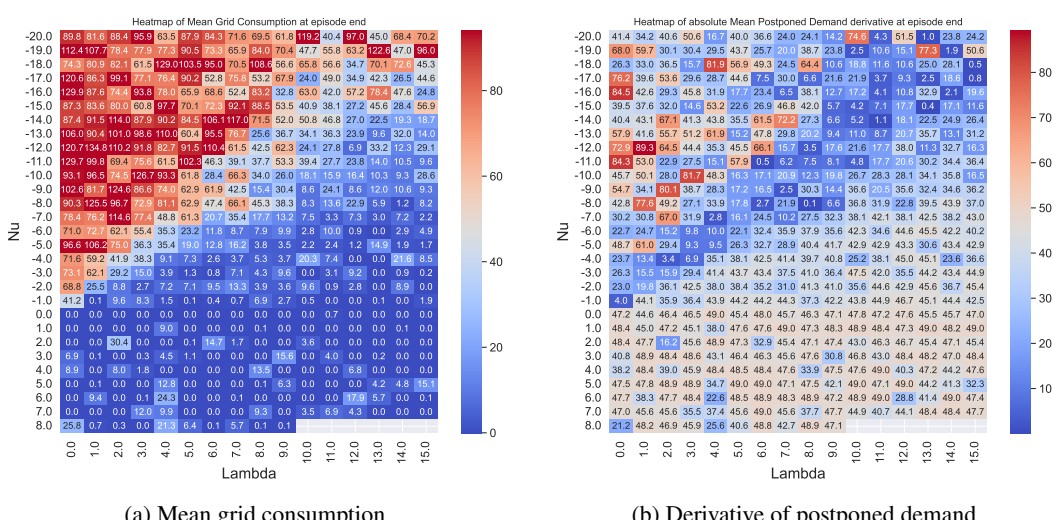

(a) Mean grid consumption          (b) Derivative of postponed demand

Figure A.1: Performance of 414 IPPO agents trained with fixed $(\lambda, \nu)$ pairs. Cells left blank correspond to diverging runs.

To benchmark our method we trained 414 independent PPO (IPPO) models, one for each point on the grid $(\lambda, \nu) \in [0, 15] \times [-20, 8]$. Figure A.1 reports, for every run, the episode-end *(a)* mean grid consumption and *(b)* absolute rate of change of cumulative postponed demand. A small derivative indicates that demand has stabilized. Only the eight multiplier pairs in Table 1 achieve both low grid consumption and vanishing postponed demand.

Table 1: Fixed multipliers $(\lambda, \nu)$ that yield stable behaviour in the IPPO grid search.

| $\lambda$ | $\nu$ |
|---|---|
| 6 | $-11$ |
| 8 | $-8$ |
| 11 | $-14$ |
| 13 | $-15$ |
| 13 | $-20$ |
| 14 | $-19$ |
| 15 | $-17$ |
| 15 | $-18$ |

Our algorithm, by contrast, identifies suitable multipliers *automatically* at execution time and maintains feasibility throughout deployment, thus avoiding expensive hyper-parameter sweeps.

## A.2 SCALABILITY STUDY

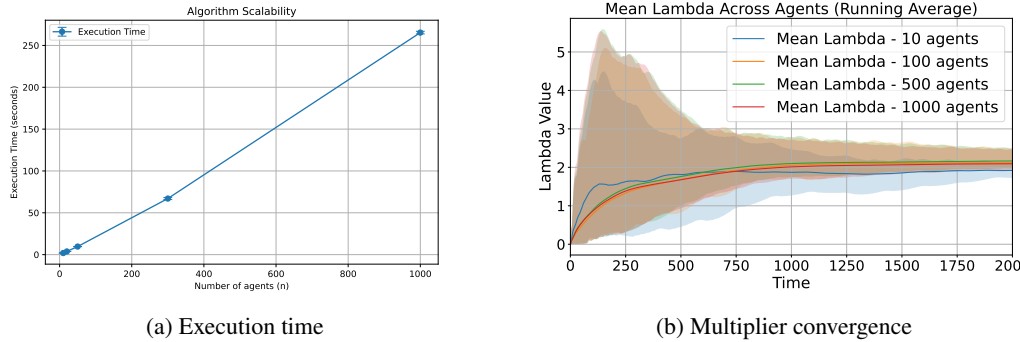

(a) Execution time        (b) Multiplier convergence

Figure A.2: Scalability of the execution phase. (a) Wall-clock execution time versus agent count. (b) Running mean of $\lambda^i$ for systems of 10, 100, 500, and 1000 agents.

Figure A.2a confirms the *linear* execution-time scaling predicted by our decentralised design, while Figure A.2b demonstrates that all agents converge to a common multiplier irrespective of population size.

## A.3 COMPARISON WITH OTHER MARL ALGORITHMS

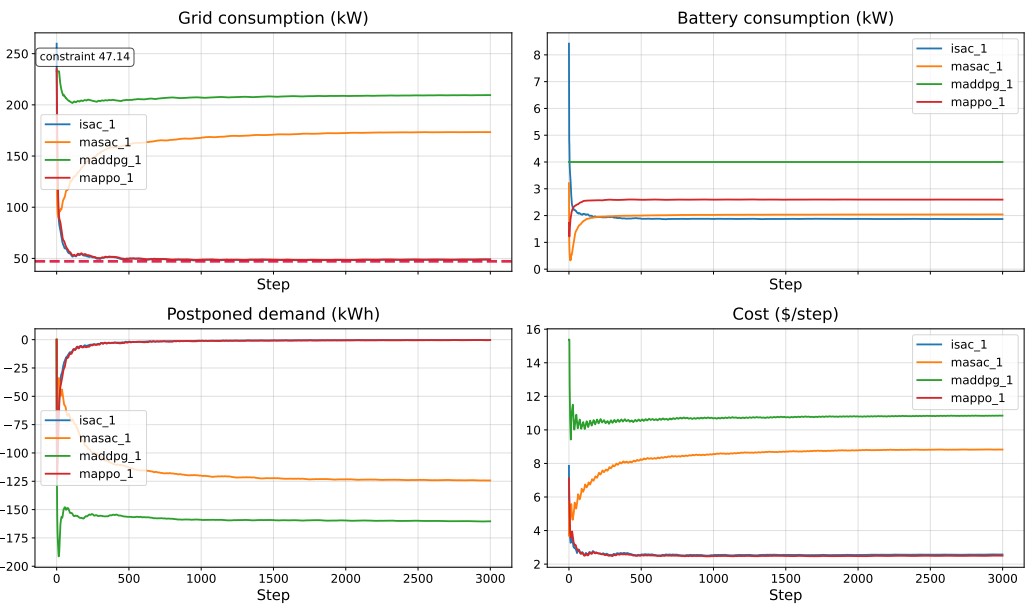

Figure A.3: Average performance of state-of-the-art MARL baselines. The grey dashed line marks the grid-consumption constraint (27 % of peak demand).

Using one of the optimal multiplier pairs from Table 1 ($(\lambda^\star, \nu^\star) = (8, -8)$), we also evaluated MAPPO, MASAC, MADDPG, and ISAC. Only ISAC and MAPPO reduce grid consumption to the desired level, yet both incur the same cost than our consensus-based method and provide no scalability guarantees.

# B  THEORETICAL ANALYSIS

## B.1  CONVERGENCE OF THE CONSENSUS ALGORITHM

In this section, we provide a rigorous analysis of the convergence properties of the consensus algorithm employed in our distributed optimization framework. The convergence properties of consensus algorithms over networks are well-studied in the literature (Olfati-Saber & Murray, 2004; Olfati-Saber et al., 2007; Xiao & Boyd, 2003). Our analysis follows standard techniques in distributed optimization and consensus algorithms, as well as properties of graph Laplacians and their spectra (Chung, 1997). Specifically, we leverage results from spectral graph theory and matrix analysis to establish the exponential convergence of our algorithm. We examine how the local dual variables $\lambda^i$ converge to a consensus value, ensuring coordination among agents while satisfying global constraints.

### B.1.1  CONSENSUS UPDATE RULE

The consensus update for agent $i$ at iteration $\ell$ can be written in a standard form for consensus algorithms:

$$
\begin{aligned}
\lambda_{\ell+1}^i &= \lambda_\ell^i - \epsilon \left( \lambda_\ell^i - \frac{1}{|\mathcal{N}^i|} \sum_{j \in \mathcal{N}^i} \lambda_\ell^j \right), \\
&= \lambda_\ell^i - \epsilon \left( \frac{1}{|\mathcal{N}^i|} \sum_{j \in \mathcal{N}^i} (\lambda_\ell^i - \lambda_\ell^j) \right), \\
&= \lambda_\ell^i - \epsilon \sum_{j \in \mathcal{N}^i} \frac{1}{|\mathcal{N}^i|} \left( \lambda_\ell^i - \lambda_\ell^j \right).
\end{aligned} \tag{B.1}
$$

where $\epsilon > 0$ is the consensus step size. This update rule adjusts each agent's dual variable towards the average of its neighbors' dual variables.

### B.1.2  MATRIX FORMULATION

We consider a communication network among the agents, given by an undirected graph $G = (V, E)$, where $V$ is the set of vertices (agents) and $E \subset V \times V$ is the set of edges (communication links between agents). The neighborhood of a node $i \in V$, denoted by $\mathcal{N}^i$, is the set of nodes that are directly connected to node $i$ by an edge; i.e., $\mathcal{N}^i = \{j \in V \mid (i,j) \in E\}$.

We aim to express the collective updates in matrix form to facilitate the convergence analysis. To do this, we first define the necessary matrices and vectors.

Let $\lambda_\ell = [\lambda_\ell^1, \lambda_\ell^2, \ldots, \lambda_\ell^N]^T \in \mathbb{R}^N$ be the global vector of local dual variables at iteration $\ell$, and let $\mathbf{1} \in \mathbb{R}^N$ be a vector of ones. We denote by $A \in \mathbb{R}^{N \times N}$ the adjacency matrix of the graph, where

$$
A(i,j) = \begin{cases} 1, & \text{if } (i,j) \in E, \\ 0, & \text{otherwise,} \end{cases} \tag{B.2}
$$

and by $D \in \mathbb{R}^{N \times N}$ the diagonal degree matrix with $D(i,i) = |\mathcal{N}^i|$. We define the (unnormalized) graph Laplacian as $L = D - A$, and the random-walk normalized Laplacian as $L^{\text{rw}} = D^{-1} L = I - D^{-1} A$.

Then the update of $\lambda_{\ell+1}$ in vector form is:

$$
\begin{aligned}
\lambda_{\ell+1} &= \lambda_\ell - \epsilon \left( \lambda_\ell - D^{-1} A \lambda_\ell \right), \\
&= \lambda_\ell - \epsilon L^{\text{rw}} \lambda_\ell, \\
&= P \lambda_\ell,
\end{aligned} \tag{B.3}
$$

where $P = I - \epsilon L^{\text{rw}}$ is the *Perron matrix*, and $I$ is the identity matrix. The graph Laplacian $L^{\text{rw}}$ captures the connectivity of the communication network among agents.

### B.1.3 ASSUMPTIONS FOR CONVERGENCE

To analyze the convergence of the consensus algorithm, we make the following assumptions:

**Assumption B.1** (Connected Graph). The communication graph $G = (V, E)$ is undirected and connected; that is, there exists a path between any pair of agents.

**Assumption B.2** (Step Size). The consensus step size $\epsilon$ satisfies $0 < \epsilon < 1$, ensuring that $P$ remains a stochastic matrix with non-negative entries.

Assumption B.1 ensures that information can propagate through the network, which is necessary for achieving global consensus. Assumption B.2 provides a bound on the step size to guarantee convergence.

### B.1.4 CONVERGENCE ANALYSIS

We analyze the convergence of the consensus algorithm by examining the properties of the Perron matrix $P$.

**Lemma B.3** (Properties of the Perron Matrix). *Under Assumptions B.1 and B.2, the Perron matrix $P = I - \epsilon L^{rw}$ satisfies the following properties:*
*(a) $P$ is row-stochastic and irreducible.*
*(b) The eigenvalues of $P$ are $\nu_i = 1 - \epsilon \Lambda_i$, where $\Lambda_i$ are the eigenvalues of the Laplacian $L^{rw}$.*
*(c) All eigenvalues of $P$ satisfy $|\nu_i| \leq 1$, the eigenvalue $\nu_1 = 1$ has algebraic multiplicity one, and all other eigenvalues satisfy $|\nu_i| < 1$.*

*Proof.* **(a) Row-Stochasticity and Irreducibility:** The elements of $P$ are given by

$$P(i,j) = \begin{cases} 1 - \epsilon, & \text{if } i = j, \\ \frac{\epsilon}{|\mathcal{N}^i|}, & \text{if } (i,j) \in E, \\ 0, & \text{otherwise.} \end{cases}$$

For each row $i$, the sum of the entries is

$$\begin{aligned} \sum_{j=1}^{N} P(i,j) &= P(i,i) + \sum_{j \in \mathcal{N}^i} P(i,j) \\ &= (1 - \epsilon) + \sum_{j \in \mathcal{N}^i} \frac{\epsilon}{|\mathcal{N}^i|} \\ &= (1 - \epsilon) + \epsilon \cdot \frac{|\mathcal{N}^i|}{|\mathcal{N}^i|} \\ &= (1 - \epsilon) + \epsilon = 1. \end{aligned}$$

Thus, $P$ is row-stochastic. Since the graph $G$ is connected (Assumption B.1), and $P$ is non-negative, it follows that $P$ is irreducible.

**(b) Eigenvalues of $P$:** Let $\Lambda_i$ be the eigenvalues of $L^{rw}$ with corresponding eigenvectors $v_i$. Then,

$$L^{rw} v_i = \Lambda_i v_i.$$

Therefore,
$$P v_i = (I - \epsilon L^{rw}) v_i = v_i - \epsilon L^{rw} v_i = v_i - \epsilon \Lambda_i v_i = (1 - \epsilon \Lambda_i) v_i.$$

Thus, the eigenvalues of $P$ are $\nu_i = 1 - \epsilon \Lambda_i$.

**(c) Eigenvalues within $[-1, 1]$:** Since the random-walk Laplacian $L^{rw}$ is similar to the symmetric normalized Laplacian

$$L^{sym} = D^{-\frac{1}{2}} L D^{-\frac{1}{2}}$$

via

$$L^{sym} = D^{\frac{1}{2}} L^{rw} D^{-\frac{1}{2}},$$

they share the same set of eigenvalues $\{\Lambda_i\}$.

To see this more explicitly, let $\Lambda_i$ and $v_i$ be an eigenvalue–eigenvector pair of $L^{\mathrm{sym}}$, i.e.,

$$L^{\mathrm{sym}} v_i \;=\; \Lambda_i v_i \quad \Longleftrightarrow \quad \left(I \;-\; D^{-\frac{1}{2}} A \, D^{-\frac{1}{2}}\right) v_i \;=\; \Lambda_i v_i.$$

Pre-multiplying both sides by $D^{-\frac{1}{2}}$ gives

$$\left(D^{-\frac{1}{2}} \;-\; D^{-1} A \, D^{-\frac{1}{2}}\right) v_i \;=\; \Lambda_i \, D^{-\frac{1}{2}} v_i \quad \Longleftrightarrow \quad \left(I \;-\; D^{-1} A\right) D^{-\frac{1}{2}} v_i \;=\; \Lambda_i \, D^{-\frac{1}{2}} v_i.$$

Recalling that $L^{\mathrm{rw}} = I - D^{-1} A$, it follows that

$$L^{\mathrm{rw}} \left(D^{-\frac{1}{2}} v_i\right) \;=\; \Lambda_i \left(D^{-\frac{1}{2}} v_i\right).$$

Hence, if $(\Lambda_i, v_i)$ is an eigenvalue–eigenvector pair of $L^{\mathrm{sym}}$, then the same $\Lambda_i$ and $D^{-\frac{1}{2}} v_i$ form an eigenvalue–eigenvector pair of $L^{\mathrm{rw}}$. Therefore, both matrices share the same eigenvalues. We can establish the following facts:

1. **Real symmetry and positive semidefiniteness:** The matrix $L^{\mathrm{sym}}$ is real symmetric (since $L$ is symmetric and $D^{-1/2}$ is diagonal). Then, $L^{\mathrm{sym}}$ is diagonalizable, and its eigenvalues are real. Standard results in spectral graph theory further show $L^{\mathrm{sym}}$ is positive semidefinite, implying its eigenvalues are nonnegative (Chung, 1997; Godsil & Royle, 2001).

2. **Eigenvalues in $[0, 2]$:** From classical bounds on the spectrum of $L^{\mathrm{sym}}$ (e.g., using the structure of the degree and adjacency matrices), one obtains

$$0 = \Lambda_1 \leq \Lambda_2 \leq \cdots \leq \Lambda_N \leq 2 \quad \text{(Horn \& Johnson, 2012; Mohar et al., 1991).}$$

   The eigenvalues of $L^{\mathrm{rw}}$ also lie in $[0, 2]$.

Because $0 \leq \Lambda_i \leq 2$ and $0 < \epsilon < 1$, we have

$$|\nu_i| = |1 - \epsilon \Lambda_i| \leq 1,$$

Since $G$ is connected, the multiplicity of the zero eigenvalue of $L^{\mathrm{rw}}$ is one, so the eigenvalue $\nu_1 = 1$ of $P$ has algebraic multiplicity one. Since there are no complex eigenvalues ($L^{sym}$ is real and symmetric), all other eigenvalues satisfy $|\nu_i| < 1$ for $i \geq 2$. Hence, all eigenvalues of $P$ lie in $[-1, 1]$.

This ensures that the spectral radius of $P$ is $\rho(P) = 1$, and the convergence of the consensus algorithm is governed by the second-largest eigenvalue in magnitude, which is less than 1. $\qquad\square$

## B.2 GLOBAL CONSENSUS ERROR

We analyze the convergence of the consensus algorithm by first establishing the value to which it converges, and then proving the rate of convergence.

### B.2.1 CONSENSUS VALUE

**Lemma B.4** (Consensus Value). *Under Assumption B.1, the consensus algorithm converges to a weighted average of the initial dual variables. Specifically, for any initial vector $\lambda_0 \in \mathbb{R}^N$,*

$$\lim_{\ell \to \infty} \lambda_\ell = \hat{\lambda} \mathbf{1},$$

*where*

$$\hat{\lambda} = \sum_{i=1}^{N} w^i \lambda_0^i,$$

*and the weights $w^i$ are given by*

$$w^i = \frac{|\mathcal{N}^i|}{\sum_{j=1}^{N} |\mathcal{N}^j|}.$$

*Proof.* By the Perron-Frobenius theorem (Horn & Johnson, 2012), since $P$ is a primitive nonnegative matrix, it satisfies

$$\lim_{\ell \to \infty} P^\ell = \mathbf{v}_1 \mathbf{w}_1^\top,$$

where $\mathbf{v}_1 = \mathbf{1}$ is the right eigenvector corresponding to the eigenvalue 1, and $\mathbf{w}_1$ is the unique left eigenvector satisfying $\mathbf{w}_1^\top P = \mathbf{w}_1^\top$ with $\mathbf{v}_1^\top \mathbf{w}_1 = 1$.

The consensus iteration is given by

$$\lambda_\ell = P^\ell \lambda_0. \tag{B.4}$$

Taking the limit as $\ell \to \infty$,

$$\lim_{\ell \to \infty} \lambda_\ell = \lim_{\ell \to \infty} P^\ell \lambda_0 = \mathbf{v}_1 \mathbf{w}_1^\top \lambda_0 = \mathbf{1}(\mathbf{w}_1^\top \lambda_0) = \hat{\lambda}\mathbf{1}.$$

This shows that all agents' dual variables converge to the scalar $\hat{\lambda}$, which is a weighted average of the initial values.

To explicitly determine $\mathbf{w}_1$, consider the transition matrix $P^{\text{rw}} = D^{-1}A$, associated with the random walk normalized Laplacian $L^{rw}$ where $A$ is the adjacency matrix and $D$ is the degree matrix. For an undirected graph, $P^{\text{rw}}$ satisfies the detailed balance condition (Levin et al., 2009):

$$w^i P_{ij}^{\text{rw}} = w^j P_{ji}^{\text{rw}}.$$

Substituting $P^{rw}(i,j) = \frac{A(i,j)}{|\mathcal{N}^i|}$ and $P^{rw}(j,i) = \frac{A(j,i)}{|\mathcal{N}^j|}$, and since $A(i,j) = A(j,i)$ for undirected graphs, we obtain

$$\frac{w^i}{|\mathcal{N}^i|} = \frac{w^j}{|\mathcal{N}^j|} = c,$$

$$w^i = c|\mathcal{N}^i|$$

Since $\sum_{i=1}^N w^i = 1$, this implies that

$$c = \frac{1}{\sum_{i=1}^N |\mathcal{N}^i|},$$

Therefore, the consensus value is

$$\hat{\lambda} = \sum_{i=1}^N w^i \lambda_0^i = \frac{\sum_{i=1}^N |\mathcal{N}^i| \lambda_0^i}{\sum_{i=1}^N |\mathcal{N}^i|},$$

which is the degree-weighted average of the initial dual variables. $\qquad\square$

### B.2.2 CONVERGENCE RATE

We now establish the exponential convergence rate to the consensus value $\hat{\lambda}$.

**Theorem B.5** (Exponential Convergence to Consensus)**.** *Under Assumptions B.1 and B.2, the consensus algorithm converges exponentially fast to $\hat{\lambda}\mathbf{1}$. Specifically, for any initial vector $\lambda_0 \in \mathbb{R}^N$,*

$$\|\lambda_\ell - \hat{\lambda}\mathbf{1}\| \leq C\rho^\ell \|\lambda_0 - \hat{\lambda}\mathbf{1}\|, \tag{B.5}$$

*where:*

- $\rho = \max_{i \geq 2} |\nu_i| < 1$ *where $\nu_i$ are the eigenvalues of P.*

- $C = \kappa(V) = \|V\|\|V^{-1}\|$ *is the condition number of the eigenvector matrix $V$.*

*Proof.* Define the error vector at iteration $\ell$ as:

$$e_\ell = \lambda_\ell - \hat{\lambda}\mathbf{1}.$$

From Lemma B.4, we have $\lim_{\ell \to \infty} e_\ell = \mathbf{0}$.

The consensus update rule is:
$$\lambda_{\ell+1} = P\lambda_\ell,$$
which implies:
$$e_{\ell+1} = Pe_\ell.$$
Iterating this, we obtain:
$$e_\ell = P^\ell e_0.$$

Since $P$ is diagonalizable, we can express it as:
$$P = V\Gamma V^{-1},$$
where:

- $V$ is the matrix of right eigenvectors of $P$.

- $\Gamma = \mathrm{diag}(\nu_1, \nu_2, \ldots, \nu_N)$ contains the eigenvalues of $P$, with $\nu_i = 1 - \epsilon\Lambda_i$.

Substituting into the error expression:
$$e_\ell = V\Gamma^\ell V^{-1}e_0. \tag{B.6}$$

In the degree-weighted consensus setting, for eigenvalue 1, the matrix $P$ has $\mathbf{1}$ as its right eigenvector, while its left eigenvector is $\mathbf{w}_1$. By definition, the initial error is $e_0 = \lambda_0 - \hat{\lambda}\mathbf{1}$ (where $\hat{\lambda}$ is the weighted average), we then have

$$\mathbf{w}_1^\top e_0 = \sum_{i=1}^{N} w^i(\lambda_0^i - \hat{\lambda}) = 0.$$

Thus, $e_0$ lies in the subspace orthogonal to $\mathbf{w}_1$. Since $e_\ell = P^\ell e_0$ and $\mathbf{w}_1^\top P = \mathbf{w}_1^\top$, it follows that

$$\mathbf{w}_1^\top e_\ell = \mathbf{w}_1^\top P^\ell e_0 = \mathbf{w}_1^\top e_0 = 0 \quad \text{for all } \ell.$$

Hence, the error remains in the subspace orthogonal to $\mathbf{w}_1$ at every iteration, allowing us to exclude the dominant component in the consensus convergence analysis. Thus,

$$e_\ell = V_{\mathrm{red}}\Gamma_{\mathrm{red}}^\ell V_{\mathrm{red}}^{-1}e_0,$$

where:

- $V_{\mathrm{red}}$ consists of eigenvectors corresponding to $\nu_i$ for $i \geq 2$.

- $\Gamma_{\mathrm{red}} = \mathrm{diag}(\nu_2, \nu_3, \ldots, \nu_N)$.

To bound the norm of the error, we apply the sub-multiplicative property of matrix norms:
$$\|e_\ell\| \leq \|V_{\mathrm{red}}\|\|\Gamma_{\mathrm{red}}^\ell\|\|V_{\mathrm{red}}^{-1}\|\|e_0\|.$$

Since $\|\Gamma_{\mathrm{red}}^\ell\|_2 = \rho^\ell$, where $\rho = \max_{i \geq 2}|\nu_i|$, we have:
$$\|e_\ell\| \leq \|V\|\|V^{-1}\|\rho^\ell\|e_0\| = C\rho^\ell\|e_0\|,$$
where $C = \kappa(V) = \|V\|\|V^{-1}\|$ is the condition number of $V$.

Since $\rho < 1$, the error decays exponentially:
$$\|e_\ell\| \leq C\rho^\ell\|e_0\|,$$

confirming that the consensus algorithm converges exponentially fast to $\hat{\lambda}\mathbf{1}$. $\qquad\square$

### B.2.3 BOUNDING THE GLOBAL CONSENSUS ERROR

We aim to bound the consensus error $e_{k+1} = \lambda_{k+1} - \hat{\lambda}_{k+1}\mathbf{1}$, where $\hat{\lambda}_{k+1}$ is the weighted average of $\lambda_{k+1}^i$.

**Lemma B.6** (Consensus Error Recursion). *Under Assumptions B.1 and B.2, the magnitude of the consensus error satisfies the recursion*

$$\|e_{k+1}\| \leq \left\|P^{\mathscr{L}}\right\| \left\|(e_k + \alpha \Delta V_{1,k})\right\|, \tag{B.7}$$

*where $\Delta V_{1,k} = V_{1,k} - \hat{V}_{1,k}\mathbf{1}$ and $\hat{V}_{1,k} = \frac{\sum_{i=1}^{N} |\mathcal{N}^i| V_{1,k}^i}{\sum_{i=1}^{N} |\mathcal{N}^i|}$.*

*Proof.* From the gradient descent step,

$$\lambda_{k+\frac{1}{2}}^i = \left[\lambda_k^i - \alpha \left(\frac{c}{N} - V_{1,k}^i\right)\right]_+.$$

The weighted average is

$$\hat{\lambda}_{k+\frac{1}{2}} = \sum_{i=1}^{N} w^i \lambda_{k+\frac{1}{2}}^i.$$

$$= \sum_{i=1}^{N} w^i \left[\lambda_k^i - \alpha \left(\frac{c}{N} - V_{1,k}^i\right)\right]_+.$$

Using the non-expansive property of the projection $[\cdot]_+$, we can write the magnitude of the error after the gradient step as

$$\left\|e_{k+\frac{1}{2}}^i\right\| = \left\|\lambda_{k+\frac{1}{2}}^i - \hat{\lambda}_{k+\frac{1}{2}}\right\| \tag{B.8}$$

$$= \left\|\left[\lambda_k^i - \alpha \left(\frac{c}{N} - V_{1,k}^i\right)\right]_+ - \sum_{i=1}^{N} w^i \left[\lambda_k^i - \alpha \left(\frac{c}{N} - V_{1,k}^i\right)\right]_+\right\|,$$

$$\leq \left\|\lambda_k^i - \alpha \left(\frac{c}{N} - V_{1,k}^i\right) - \sum_{i=1}^{N} w^i \left(\lambda_k^i - \alpha \left(\frac{c}{N} - V_{1,k}^i\right)\right)\right\|,$$

$$= \left\|\lambda_k^i - \alpha \left(\frac{c}{N} - V_{1,k}^i\right) - \left(\hat{\lambda}_k - \alpha \left(\frac{c}{N} - \hat{V}_{1,k}\right)\right)\right\|,$$

$$= \left\|e_k^i + \alpha \left(V_{1,k}^i - \hat{V}_{1,k}\right)\right\|. \tag{B.9}$$

After the consensus update (B.4),

$$\|e_{k+1}\| \leq \left\|P^{\mathscr{L}}\right\| \left\|e_{k+\frac{1}{2}}\right\|. \tag{B.10}$$

since the consensus step only affects the error term through multiplication by $P$. Substituting (B.9) into (B.10) and letting $V_{1,k}^i - \hat{V}_{1,k} = \Delta V_{1,k}$, we obtain (B.7). $\qquad\square$

**Theorem B.7** (Asymptotic Bound on Consensus Error). *For the standard assumption of bounded rewards, the constraint functions $V_{1,k}^i$ are bounded such that $\|\Delta V_{1,k}\| \leq \sigma$ for some $\sigma > 0$. Then, the consensus error satisfies*

$$\lim_{k \to \infty} \|e_{k+1}\| \leq \frac{\rho^{\mathscr{L}} \alpha \sigma}{1 - \rho^{\mathscr{L}}}.$$

*where $\rho = 1 - \epsilon \Lambda_2$ as before.*

*Proof.* Using Lemma B.6 and Theorem B.5, we have

$$\|e_{k+1}\| \leq \left\|P^{\mathscr{L}}\right\| (\|e_k\| + \alpha\|\Delta V_{1,k}\|)$$

$$\leq \rho^{\mathscr{L}} \|e_k\| + \rho^{\mathscr{L}} \alpha \sigma, \tag{B.11}$$

since $\|P^{\mathscr{L}}\| = \rho^{\mathscr{L}}$ in the subspace orthogonal to $\mathbf{1}$.

Unrolling the recursion:

$$\begin{aligned}
\|e_{k+1}\| &\leq \rho^{\mathscr{L}}\|e_k\| + \rho\alpha\sigma \\
&\leq \rho^{2\mathscr{L}}\|e_{k-1}\| + \rho^{2\mathscr{L}}\alpha\sigma + \rho^{\mathscr{L}}\alpha\sigma \\
&\leq \dots \\
&\leq \rho^{\mathscr{L}(k+1)}\|e_0\| + \rho^{\mathscr{L}}\alpha\sigma\sum_{t=0}^{k}\rho^{t\mathscr{L}} \\
&= \rho^{\mathscr{L}(k+1)}\|e_0\| + \rho^{\mathscr{L}}\alpha\sigma\left(\frac{1-\rho^{\mathscr{L}(k+1)}}{1-\rho^{\mathscr{L}}}\right).
\end{aligned}$$

Taking the limit as $k \to \infty$, we obtain

$$\lim_{k\to\infty}\|e_{k+1}\| \leq \frac{\rho^{\mathscr{L}}\alpha\sigma}{1-\rho^{\mathscr{L}}}.$$

$\square$

