# OpenReview forum: "Scalable Constrained Multi-Agent Reinforcement Learning via State Augmentation and Consensus for Separable Dynamics"
_ICLR.cc/2026/Conference — ICLR 2026 Conference Withdrawn Submission_

### Official Review · Reviewer_qyZg · 2025-10-29

**Soundness:** 3
**Presentation:** 3
**Contribution:** 1
**Rating:** 4
**Confidence:** 2

**Summary:**

The authors propose a distributed approach for constrained multi-agent reinforcement learning (MARL), combining local policy learning with augmented state representations and distributed coordination of dual variables via consensus. The method assumes that the underlying MDP admits a factored structure, where interactions among agents occur only through a shared global resource constraint. The authors provide empirical evaluation across several environments.

**Strengths:**

The paper provides a clear formulation of the constrained MARL setting and integrates consensus-based dual variable updates in a principled way.

Theoretical analysis is included, establishing convergence bounds on the consensus error.

The experimental results demonstrate favorable scaling behavior, with performance maintained even when scaling to thousands of agents.

**Weaknesses:**

The assumption of a factorizable MDP with no coupling beyond a global constraint is quite strong. This restricts the applicability of the method to settings where agent interactions are essentially decoupled, which significantly simplifies the multi-agent problem. As a result, the method may not address non-trivial MARL scenarios with rich inter-agent dependencies or strategic interactions.

**Questions:**

Could you clarify in which settings constrained MARL is genuinely necessary? In the power grid example, it appears that the global constraint could be enforced through a centralized decision agent—for instance, by adjusting the energy price—thus removing the need for distributed multi-agent coordination.

---

### Official Review · Reviewer_DCVv · 2025-10-30

**Soundness:** 4
**Presentation:** 4
**Contribution:** 3
**Rating:** 4
**Confidence:** 4

**Summary:**

This paper presents a distributed approach to constrained multi-agent reinforcement learning (CMARL) that scales linearly with the number of agents by exploiting separable dynamics and summable rewards.
The proposed framework, called CMARL with state augmentation and consensus, integrates constrained single-agent reinforcement learning via state augmentation with a distributed consensus mechanism over Lagrange multipliers to ensure satisfaction of global constraints without central coordination.

**Strengths:**

Originality:

The integration of state-augmented constrained RL with distributed consensus over dual variables is a novel and elegant combination.
Focus on average constraint satisfaction rather than per-step safety is well-justified for resource management applications and distinguishes the work from prior safe MARL methods.

Technical Quality:

The derivation from centralized constrained MARL to a fully distributed primal–dual consensus formulation is mathematically sound.
The convergence bound is clearly stated, with appropriate assumptions on network connectivity and step sizes.
The methodology extends well-established CRL and distributed optimization frameworks into the multi-agent domain coherently.

Clarity:

The exposition is clear and well-organized, with the problem setup, assumptions, and algorithm steps explicitly stated.
Algorithm 1 provides a concise and readable description of the distributed update mechanism.

Significance:

The approach demonstrates linear scalability in both training and execution, a key advantage over CTDE-based methods.

**Weaknesses:**

The strong assumptions of separable dynamics, independent policies, and summable rewards limit applicability to a narrow class of problems.

The experiments focus almost exclusively on smart grid demand response.


There is limited quantitative exploration of communication topology effects.

No explicit analysis of how consensus step size or number of consensus rounds affects convergence and communication overhead, beyond a qualitative discussion.

**Questions:**

Can the proposed method be adapted to settings with partially coupled dynamics?

Could the consensus mechanism be extended to handle nonlinear or time-varying constraints?

How does communication delay or message loss affect convergence?

Can you include results for decentralized safe MARL baselines to highlight the trade-off between safety and scalability?

Would dynamic consensus averaging further improve performance?

---

### Official Review · Reviewer_LWno · 2025-10-31

**Soundness:** 2
**Presentation:** 2
**Contribution:** 1
**Rating:** 2
**Confidence:** 4

**Summary:**

This paper presents a novel distributed approach for constrained Multi-Agent Reinforcement Learning (MARL) that effectively scales to large numbers of agents while ensuring constraint satisfaction. The authors introduce a method that combines state augmentation with distributed consensus over Lagrange multipliers, addressing a specific class of problems where agents have separable dynamics and local observations but need to collectively satisfy constraints on global resources. The key contributions include the development of a distributed CMARL algorithm that ensures consensus and constraint satisfaction, the demonstration of linear scalability in both training and execution, and the validation of the method through experiments on smart grid management. The results show that the proposed consensus mechanism is critical for feasibility, as it enables agents to coordinate their actions to meet global constraints without centralized control. The paper also provides theoretical analysis and proofs for the convergence of the consensus algorithm, further supporting the robustness and effectiveness of the proposed approach.

**Strengths:**

1. The Problem Formulation section clearly describes the assumptions of this work and the relevant real-world domains.
2. The authors' writing in the methodology section is relatively clear.

**Weaknesses:**

1. Introduction Section：The motivation of this paper is not clearly stated. Although there is a brief description towards the end of the paper, it is vague. I suggest the authors enhance the description of the motivation to better highlight the contributions of this work.

2. Related Work Section：The related work section lacks a comprehensive review of relevant literature, including [1], [2], and [3]. After reading the entire paper, I still do not fully understand the differences between this work and the existing work. In the “Decentralized Constrained MARL”, the authors mention: “we target average constraint satisfaction rather than per-step safety, and assume separable dynamics to achieve linear scaling.” I am unclear why the original constrained methods [1] cannot directly solve this problem. I suggest the authors clarify these vague statements.

3. Method Section：There are some parts in the method section that I find hard to understand. Like Remark 4.1: “Requiring only single-scalar neighbor communication.” I am not sure what this means and why it is necessary.

4. Experiments Section：① The current experiments are too few, which will limit the contributions of this paper. I suggest that the authors supplement more experiments to fully illustrate the contributions of this paper. For example, benchmarks such as SUMO [4] or Powergrid [5]. ② I suggest that the authors should demonstrate the related work mentioned in the related work section. I see that the authors have tried to compare methods such as MAPPO and MADDPG in the experiments, but this paper emphasizes the Scalable characteristic. Comparing only these is far from enough. At least some methods like MARL with networked [2] and some methods like constraint MARL with networked [1][6] should be added.

**Reference**

[1] Ying D, Zhang Y, Ding Y, et al. Scalable primal-dual actor-critic method for safe multi-agent rl with general utilities. In NeurlPS 2023.

[2] Qu G, Lin Y, Wierman A, et al. Scalable multi-agent reinforcement learning for networked systems with average reward. In NeurlPS 2020.

[3] Feng J, Shi Y, Qu G, et al. Stability constrained reinforcement learning for decentralized real-time voltage control. IEEE Transactions on Control of Network Systems, 2023.

[4] Chu T, Chinchali S, Katti S. Multi-agent reinforcement learning for networked system control. In ICLR 2020.

[5] Chen D, Chen K, Li Z, et al. Powernet: Multi-agent deep reinforcement learning for scalable powergrid control[J]. IEEE Transactions on Power Systems, 2021, 37(2): 1007-1017.

[6] Zhang L, Li L, Wei W, et al. Scalable constrained policy optimization for safe multi-agent reinforcement learning. In NeurIPS 2024.

**Questions:**

Please refer to the “Weakness” section for related questions.

**Details Of Ethics Concerns:**

Regarding ethical review, I have no concerns.

---

### Official Review · Reviewer_zjJD · 2025-10-31

**Soundness:** 2
**Presentation:** 3
**Contribution:** 2
**Rating:** 4
**Confidence:** 4

**Summary:**

This paper proposes a distributed training-based safe MARL algorithm under the separable dynamics setting. Theoretical convergence of the proposed algorithm is established. Simulation results on a demand response problem in smart grids demonstrate the effectiveness of the proposed algorithm.

**Strengths:**

This paper is well organized and easy to follow. The proposed algorithm is validated from both theoretical and empirical perspectives.

**Weaknesses:**

The problem considered in this paper is rather simple and cannot model most multi-agent decision-making problems in real-world scenarios. Compared with existing distributed (safe) MARL algorithms, the proposed algorithm does not need to estimate the global value function because of the idealized settings of the policy and dynamics. The communication graphs are fixed, which makes this algorithm lack robustness against link failures in practice.

**Questions:**

(1) The authors are recommended to introduce the primal step in the manuscript. What is the input to the approximator for the value function?

(2) A similar simulation environment has been found in [1], which considers a similar setting as this work. Could the authors evaluate the proposed algorithm on more challenging tasks with more agents?

Reference:

[1] Amaya-Corredor, S., Calvo-Fullana, M., & Jonsson, A. Distributed Constrained Multi-Agent Reinforcement Learning with Consensus and Networked Communication. In Seventeenth European Workshop on Reinforcement Learning.

---

### Note · Authors · 2025-11-17

I have read and agree with the venue's withdrawal policy on behalf of myself and my co-authors.